# Deep-Learning-Based Predictive Imaging Biomarker Model for EGFR Mutation Status in Non-Small Cell Lung Cancer from CT Imaging

**DOI:** 10.3390/cancers16061130

**Published:** 2024-03-12

**Authors:** Abhishek Mahajan, Vatsal Kania, Ujjwal Agarwal, Renuka Ashtekar, Shreya Shukla, Vijay Maruti Patil, Vanita Noronha, Amit Joshi, Nandini Menon, Rajiv Kumar Kaushal, Swapnil Rane, Anuradha Chougule, Suthirth Vaidya, Krishna Kaluva, Kumar Prabhash

**Affiliations:** 1Department of Imaging, The Clatterbridge Cancer Centre NHS Foundation Trust, Liverpool L7 8YA, UK; 2Faculty of Health and Life Sciences, University of Liverpool, Liverpool L7 8TX, UK; 3Department of Radiodiagnosis, Tata Memorial Hospital, Mumbai 400012, Maharashtra, India; vatsal.kania@gmail.com (V.K.); ujjwalagg8@gmail.com (U.A.); renukaashtekar1@gmail.com (R.A.); drshreyashukla@gmail.com (S.S.); 4Department of Medical Oncology, Tata Memorial Hospital, Mumbai 400012, Maharashtra, India; vijaypgi@gmail.com (V.M.P.); vanita.noronha@gmail.com (V.N.); amitjoshi74@gmail.com (A.J.); nandini.menon1412@gmail.com (N.M.); anu.c1112@gmail.com (A.C.); kprabhash1@gmail.com (K.P.); 5Department of Pathology, Tata Memorial Hospital, Mumbai 400012, Maharashtra, India; rajiv.kaushal@gmail.com (R.K.K.); raneswapnil82@gmail.com (S.R.); 6Predible Health, IKP Eden, Bangalore 560029, Karnataka, India; suthirth@predible.com (S.V.); krishna@predible.com (K.K.)

**Keywords:** non-small cell lung cancer, EGFR, semantics, radiomics, deep learning, machine learning

## Abstract

**Simple Summary:**

Deep-learning-based radiogenomic (DLR) models show promising performance in assisting with lung cancer care. The primary aim of our study was to develop and validate a DLR model to predict EGFR mutation status in non-small-cell lung cancer (NSCLC) patients. Using 990 patients from two clinical trials, the study employed a machine learning pipeline that analysed CT images with manually selected tumour regions. Two deep convolutional neural networks segmented lung masses and nodules from 3D regions of the CT image. The combined radiomics and DLR model achieved 88% accuracy in predicting EGFR mutations, outperforming individual models. The semantic features extracted from CT images also contributed to accurate predictions. The study suggests that this AI-based model in combination with CT semantic features could serve as a non-invasive biomarker that aids in predicting EGFR mutation status with significant accuracy.

**Abstract:**

Purpose: The authors aimed to develop and validate deep-learning-based radiogenomic (DLR) models and radiomic signatures to predict the EGFR mutation in patients with NSCLC, and to assess the semantic and clinical features that can contribute to detecting EGFR mutations. Methods: Using 990 patients from two NSCLC trials, we employed an end-to-end pipeline analyzing CT images without precise segmentation. Two 3D convolutional neural networks segmented lung masses and nodules. Results: The combined radiomics and DLR model achieved an AUC of 0.88 ± 0.03 in predicting EGFR mutation status, outperforming individual models. Semantic features further improved the model’s accuracy, with an AUC of 0.88 ± 0.05. CT semantic features that were found to be significantly associated with EGFR mutations were pure solid tumours with no associated ground glass component (*p* < 0.03), the absence of peripheral emphysema (*p* < 0.03), the presence of pleural retraction (*p* = 0.004), the presence of fissure attachment (*p* = 0.001), the presence of metastatic nodules in both the tumour-containing lobe (*p* = 0.001) and the non-tumour-containing lobe (*p* = 0.001), the presence of ipsilateral pleural effusion (*p* = 0.04), and average enhancement of the tumour mass above 54 HU (*p* < 0.001). Conclusions: This AI-based radiomics and DLR model demonstrated high accuracy in predicting EGFR mutation, serving as a non-invasive and user-friendly imaging biomarker for EGFR mutation status prediction.

## 1. Introduction

Non-small-cell lung cancer (NSCLC) accounts for the majority (85%) of all lung cancer cases. The two most common histopathologic subtypes are adenocarcinoma and squamous cell carcinoma [1]. In the modern era of personalized and precision medicine, the mutational testing of selected genes for NSCLC remains a standard practice to categorize patients into responders and non-responders. This includes testing for mutations of the epidermal growth factor receptor (EGFR), a cell surface receptor activating cell growth and survival, which when mutated confers sensitivity to tyrosine kinase inhibitors. Some common clinical characteristics seen in patients with EGFR mutations are non-smoking status, adenocarcinoma histology, female sex, and East Asian ethnicity [2,3].

A lung mass Trucut biopsy is a must for histological and mutational analysis seeking to further develop a plan of treatment. However, it is not always feasible to obtain adequate tissue samples from a biopsy for mutational analysis, or there might be errors in targeting the lung mass. Some high-risk patients might not be fit to undergo a Trucut biopsy due to a deranged coagulation profile or other underlying morbidities. There is always a small risk of life-threatening complications associated with biopsies such as pneumothorax, haemoptysis due to alveolar haemorrhage, haemothorax, and hemopericardium. Also, in developing countries like India, advanced laboratory facilities for genomic mutation studies are not widely available, especially in small towns and rural areas. In this situation, clinical parameters such as Asian ethnicity, female sex, non-smoking status, and adenocarcinoma histology have been considered as potential prerequisites for the presence of EGFR mutation [4,5]. However, these clinical characteristics represent only a selected small population with higher probability of harbouring the EGFR mutation. The tumour cells harvested via a core biopsy represent only a tiny fraction of the tumour, and might not represent the complex heterogeneity of the tumour mass. A study conducted by Taniguchi et al. [6] analysed 50–60 areas of tumour tissue in 21 patients with a known EGFR mutation. Intra-tumoural heterogeneity was seen in 28.6% of the study cohort (6 out of 21 tumours), which contained both EGFR-mutated and wild-type cells. Thus, more detailed factors are needed to analyse EGFR mutation statuses, such as the characterisation and analysis of quantitative computed tomography (CT) features. Every patient with a lung mass needs to undergo a CT scan; hence, pre-treatment CT images can prove to be a rich source of data for analysis. They can provide additional data for genomics and can potentially identify tumours with EGFR mutations [7,8]. The resampling of the tumour can be considered if there are discrepancies between the mutation results from the biopsy and CT findings (based on deep learning, radiomics, and semantic markers); these combined analyses can potentially reduce the chances of missing EGFR mutations in a tumour mass.

Medical imaging is intuitively very suitable as a biomarker source, especially in lung cancer patients, where it is used to visualize tumour phenotypes and predict treatment response. There has been an increase in research on the characterisation of quantitative imaging features reflecting tumour biology, physiology, and phenotype using artificial intelligence (AI)-based algorithms. Radiomics and deep-learning (DL)–AI-based models are extensively used with medical imaging [9,10,11]. Radiomics refers to the computerized extraction of data from radiologic images, and provides unique potential for making lung cancer screening more rapid and accurate by using machine learning algorithms. For analysing the tumour area, radiomics models require the precise annotation of the tumour boundary, which requires manually marking the tumour on all three planes [12,13]. Since only the tumour area is taken into consideration, the microenvironment of the surrounding lung parenchyma is ignored. Advanced AI models such as neural network-based DL methods can overcome these limitations through a self-learning strategy, and present a promising tool for genomic analysis [14,15,16,17]. The DL method can be likened to the functioning of the neural network in the brain. In comparison to radiomic methods, precise tumour boundary annotation is not required with deep learning, thus saving a lot of time and human effort. Furthermore, the DL method takes into consideration the microenvironment of the surrounding lung parenchyma, and can extract features that are adaptive to specific clinical outcomes, whereas radiomics can only describe general features that lack specificity for outcome prediction [18,19,20]. Moreover, with the help of the DL model, the sub areas within the tumour that are strongly related to EGFR mutation status can be identified and further subjected to biopsy if required. Thus, both the methods can directly or indirectly help clinicians make rapid treatment decisions for patients.

The primary purpose of this study is to develop radiomics and DL models, which can mine data from CT images to predict EGFR mutation status using a large cohort of patients with NSCLC. Our DL method is an end-to-end pipeline that requires only the manual marking of the tumour region in a CT image without precise annotation [21]. We also identified specific CT-based semantic features that correlate strongly with the presence of positive EGFR mutation in our study population.

## 2. Materials and Methods

The study was approved by the Institutional Ethics Committee and a waiver for consent was obtained in view of the retrospective nature of the study. The inclusion criteria for enrolling patients into the study were (1) primary lung adenocarcinoma confirmed on histopathology report; (2) the presence of proven records of EGFR mutation status; and (3) pre-operative/baseline contrast-enhanced CT data available. Exclusion criteria were (1) incomplete medical records or non-availability of digital DICOM CT images; (2) patients who have received chemotherapy or radiotherapy outside our institute before the baseline scan; (3) any other active illness or pathological condition that might interfere with the study data as per medical records. Finally, 990 patients were included in the study with patient cases evaluated from 2010 January to 2016 December. These patients were accrued from two clinical trials, which evaluated the role of Gefitinib vs. Pemetrexed and Carboplatin in the treatment of EGFR-mutated NSCLC [22,23].

The tumour specimens were obtained using CT-guided Trucut biopsy, with the biopsy targeted enhanced solid components of the tumour. EGFR mutations were identified on four tyrosine kinase domains (exons 18–21), which are common mutations in lung cancer. The mutation status was determined using a TaqMan Probe-Based Endpoint Genotyping Mutation Analysis undertaken via Real-Time PCR on the LC 480 II platform. For identifying a tumour as an EGFR mutant, any one exon (exon 18–21) mutation should be present; otherwise, the tumour should be identified as EGFR wild-type. The focus of the study was predicting the EGFR mutation status.

### 2.1. Radiology Review

A clinical radiologist with 10 years of experience in thoracic imaging and another radiologist with 2 years of experience in general radiology retrospectively reviewed the CT scans. Both the radiologists were blinded to clinical and histologic findings. The imaging review was performed on reconstructed DICOM data using a volume viewer integrated within the PACS. The images were reviewed for lung, soft-tissue, and bone window with reformatting available in all three planes, i.e., axial, coronal and sagittal. In case of any disagreements between the radiologists as regards the CT findings, the majority class was used as the final CT feature. Mean values were used for continuous variables. A subset of 223 patients was selected for the extraction of pre-determined semantic features from CT. The clinical details of the same subset of patients were also collected.

### 2.2. Development of the DL Model

Using a convolutional neural network, DL aims to learn the abstract mapping between the raw data and the desired label. Our DL model for EGFR mutation classification is a linear support vector machine (SVM), which takes in 9 different feature vectors extracted from 6 DL models and uses them to classify the EGFR mutation status. The 6 DL models include 2 models trained to segment masses, 2 models trained for nodule texture classification, 1 model for nodule spiculation classification, and another model for nodule segmentation. The selection of ROI for the DL model and its illustration is shown in Figure 1 and Figure 2. Table 1 shows the list of 9 feature vectors, the models used and their combinations. For each model, a patch was extracted around the largest mass in the study. The DL model was constructed using the following frameworks: Python 3.8, Pytorch 1.1.

For all the segmentation models, we used a standard 3D U-Net architecture, and a 512-dimensional feature vector was taken from the bottom-most layer of the U-Net. Figure 3 shows the architecture of a standard U-Net with 512 features at the bottom-most layer. For all the classification tasks, a standard 3D Wide ResNet was used. The feature vector was extracted from the layer before the AveragePool3D layer. Figure 4 shows the structure of a standard 3D Wide ResNet. Feature vector was extracted from the conv4 layer before the avg-pooling layer.

Both the mass segmentation networks were trained using the above 990 patient data. All the masses were annotated by a technologist and were reviewed by an experienced radiologist. For nodule segmentation, texture classification and spiculation classification networks, 1010 studies from the publicly available dataset LIDC-IDRI were used for training. All these models were trained with a learning rate of 1 × 10^−4^ and a weight decay of 1 × 10^−5^. Segmentation networks were trained with Negative-Log Likelihood (NLL) loss and classification networks were trained with Cross Entropy loss. Augmentations such as rotate, flip, scale and translate were used for all the models.

Once all the 9 feature vectors had been extracted, we generated a combined feature vector of 4032 features. Of these 4032 features, 789 feature columns were zero, leaving a feature vector of size 3243. An SVM with linear kernel and balanced weights for the 2 classes was trained on the 3242 features of 990 patients with 3-fold cross-validation. Using the coefficients of the best model obtained, we removed all the feature columns with coefficients ≤ N. We iterated on various values for N and found N = 0.04 yielded the best model. After using N = 0.04 to remove the small coefficients, we had 1422 features left. We retrain the SVM with the new set of features to generate our best model.

On the subset with semantic features, the same experiments with the same set of 9 feature vectors were conducted with and without the semantic features.

### 2.3. Development of Radiomics Model

The primary tumour was segmented using the following techniques: manual, semi-automated and automated segmentation methods. The primary tumour on contrast-enhanced CT was delineated manually using the post-processing software (AW 4.4) by a radiologist with ten years of experience in thoracic imaging. The tumour was first annotated in the mediastinal window (W 330 HU; L 50 HU) to include only the tumour area by identifying boundaries with the chest wall and other soft tissues, as shown in Figure 5, then in the lung window (W 1500 HU; L −600 HU) to delineate the maximum extent of the lung parenchyma.

The segmented image was then subjected to the extraction of radiomics features using pyradiomics [24]. A total of 1110 radiomic features were calculated, divided into five groups: tumour intensity (n = 19), texture (n = 95), wavelet (n = 912), Laplacian of Gaussian (n = 74), and shape (n = 19). Emphasis was placed on the features from the previously published prognostic radiomic signatures: (I) tumour intensity—“Energy”, (II) texture—“Gray Level Nonuniformity”, (III) wavelet—“Gray Level Nonuniformity HLH”, and (IV) shape—“Compactness”. Our radiomics model for EGFR mutation classifier is a linear SVM, which takes in the 1110 radiomic features and predicts the presence of EGFR mutation. No feature columns with all zeros were identified. An SVM with linear kernel and balanced weights for the 2 classes was trained on the 1110 features of 990 patients with 3-fold cross-validation. Using the coefficients of the best model obtained, we removed all the feature columns with coefficients ≤ N. We iterated on various values for N and found that N = 0.1 yielded the best model. After using N = 0.1 to remove the small coefficients, we had 200 features left. We retrained the SVM with the new set of features to generate our best model. Table 2 shows top-performing radiomics features in predicting EGFR mutation. Figure 6 shows the pattern of radiomic workflow. Figure 7 shows the covariance matrix of radiomic features.

### 2.4. Combining DL and Radiomic Features

We combined the 4032 features from DL models and 1110 radiomic features and trained an SVM to predict the EGFR mutation. Similar to the above methods, the coefficients of the initial SVM were used to identify the best contributing features. For this model, N = 0.05 gave the best results. At N = 0.05, we were left with the top 2000 features, which were then used to retrain the SVM to generate out best model.

### 2.5. Statistical Analysis

Statistical analysis was performed using SPSS version 21 (IBM, Armonk, NY, USA). Data were descriptively analysed using frequency and percentage for categorical data. Interobserver agreement was determined by calculating Kappa values. A chi square test for independence was used to observe if any association could be seen between two variables. The Mann–Whitney U test was used to compare the medians between the two groups. Univariate Binomial logistic regression was used to determine the predictive factors for EGFR mutation. Multiple logistic regression analyses were performed to identify independent factors that can be used to predict EGFR mutation status. The final model was selected with the backward elimination method. Area under the curve (AUC) and Receiver Operating Characteristics (ROC) curves were used to present the accuracy of different predictive models. All statistics were 2-sided, and a value of *p* < 0.05 was considered statistically significant.

## 3. Results

### 3.1. Patient’s Characteristics

The relevant clinical data of a subset of 223 subjects are given in Table 3.

### 3.2. Correlation of EGFR Mutation Status with Clinical Features

The median ages of patients did not differ between the EGFR wild-type and EGFR mutant (*p* = 0.095) (Refer Table 1). EGFR mutation rates were significantly higher (a) in women than in men (*p* < 0.001) and (b) in non-smokers than in smokers (*p* < 0.001). Statistical analysis also revealed that Stage III disease was frequently seen with EGFR wild-type tumours (*p* < 0.008).

### 3.3. Correlation of EGFR Mutation Status with Semantic Features

Semantic features were extracted for 223 patients. The semantic features have been outlined in Table 4. Of 28 CT semantic features, univariate analysis (Table 4) revealed that the following CT features were significantly associated with harbouring EGFR mutation in NSCLC patients, including (a) pure solid tumours with no associated ground glass component (*p* < 0.03), (b) the absence of peripheral emphysema (*p* < 0.03), (c) the presence of pleural retraction (*p* = 0.004), (d) the presence of fissure attachment (*p* = 0.001), (e) the presence of a metastatic nodule in both the tumour-containing lobe (*p* = 0.001) and the non-tumour-containing lobe (*p* = 0.001) (f) the presence of ipsilateral pleural effusion (*p* = 0.04) and (f) the average enhancement of the tumour mass above 54 HU (*p* < 0.001).

### 3.4. Radiomics Model Used in Predicting EGFR Mutation

The SVM model for EGFR mutation classification using radiomic features had an AUC of 0.72 ± 0.03 with three-fold cross-validation in 990 studies. Figure 8a shows the ROC curve of the model.

### 3.5. DL Model in Predicting EGFR Mutation

The SVM model for EGFR mutation classification using the nine feature vectors generated from DL models had an AUC of 0.82 ± 0.01 (CI: 0.81, 0.83) with three-fold cross-validation on 990 patients. Figure 8b shows the ROC curves for the three repetitions and their respective AUCs. On the subset of 223 cases with semantic features, the model had an AUC of 0.84 ± 0.02 (CI: 0.82, 0.86) without any semantic features, whereas the model with semantic features had an AUC of 0.88 ± 0.05 (CI: 0.83, 0.93). There was a 4% improvement in AUC following the addition of semantic features irrespective of the value of N used. Figure 8c,d show the ROC curves for the smaller subset of cases without and with semantic features added.

### 3.6. Combining DL and Radiomic Features

The SVM model for EGFR mutation classification using the nine feature vectors generated from DL models, and 1110 radiomic features had an AUC of 0.88 ± 0.03 with three-fold cross-validation on 990 studies. Figure 9 shows the ROC curve of the model.

## 4. Discussion

In this study, we assessed the role of AI-based radiomics and DL models using pre-treatment CT images of patients with lung adenocarcinoma to predict the EGFR mutation status. The DL model was trained using the CT images of 990 patients with three-fold cross-validation. The radiomics model showed good predictive performance with an AUC of 0.72 ± 0.03 with three-fold cross-validation on 990 studies. The SVM model generated from DL models had an AUC of 0.82 ± 0.01 with three-fold cross-validation on 990 studies. On a smaller subset of 223 cases for which semantic features were extracted, the DL model had an AUC of 0.84 ± 0.02, which improved to an AUC of 0.88 ± 0.05 when the semantic features were combined. There was a 4% improvement in AUC following the addition of semantic features. Cases with marked heterogeneity in the tumour, in cases with large tumour sizes and associated collapse or consolidation, showed the reduced accuracy of the DL and radiomics model. This resulted in increased error in the manual as well as automatic annotation of tumours, including the annotation of non-tumoural segment, and by extension, errors in feature extraction and texture analysis. With increasing tumour size and heterogeneity, there is a loss in the internal characteristics of tumours related to specific genetic mutation, which results in the inaccurate training of the model and a reduction in predictive performance.

Other similar studies have shown a similar utility of DL models with improvements in the AUC when combined with clinical parameters [25,26,27,28]. Further, a study on the PET/CT fusion algorithm using a dataset of 150 patients showed a prediction accuracy of EGFR and non-EGFR mutations of 86.25% in the training dataset and 81.92% in the validation set [29].

Clinical utility of the DL model: Our analysis provides an alternative effective method to assess EGFR mutation in patients with NSCLC without requiring any intervention. It can also act as an effective supplement to a biopsy. It can also help avoid complications associated with biopsies, and reduce false negative biopsy results due to tumour heterogeneity. In such cases, if the deep learning model shows a high probability of EGFR mutation, re-biopsy can be attempted. The DL method can further assist in selecting the target area for biopsy. Since the human assistance required is minimal, a large amount of data can be processed with minimal errors and time. The model is easy to use and apply at various levels of healthcare settings. The deep learning model only requires routinely used CT images, without adding any extra cost. Therefore, this model can be used multiple times throughout the course of treatment.

Limitations: The study was conducted on a population from a single tertiary healthcare centre. The model needs to be further trained and validated on large multicentric cohorts to increase the accuracy and robustness. In the current study, only EGFR mutation status was taken into consideration. The relationship between EGFR mutation and other genetic mutations (e.g., ROS-1, ALK) can be explored in future work, as has been explored in a few other preliminary studies [30].

## 5. Conclusions

Radiomics and deep learning models show promising results in the prediction of EGFR mutation status. The accuracy is further increased when CT semantic features are taken into consideration, along with the deep learning model. The application of both the models in clinical practice can be useful in predicting EGFR mutation status in a patient while the lung biopsy or genetic mutation test results are still being awaited. Further improvements in the sensitivity and specificity of both the models are expected with larger data sets.

## Figures and Tables

**Figure 1 cancers-16-01130-f001:**
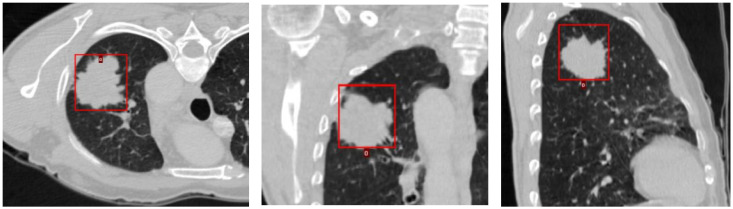
Selection of ROI (red box) for DL model.

**Figure 2 cancers-16-01130-f002:**
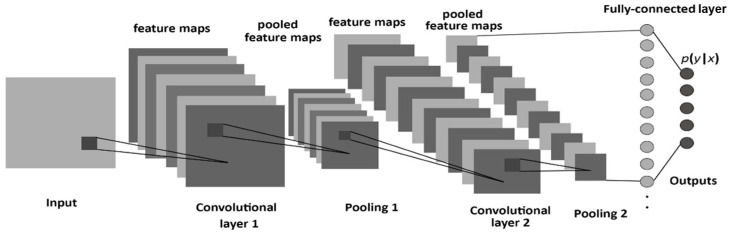
Illustration of the DL model.

**Figure 3 cancers-16-01130-f003:**
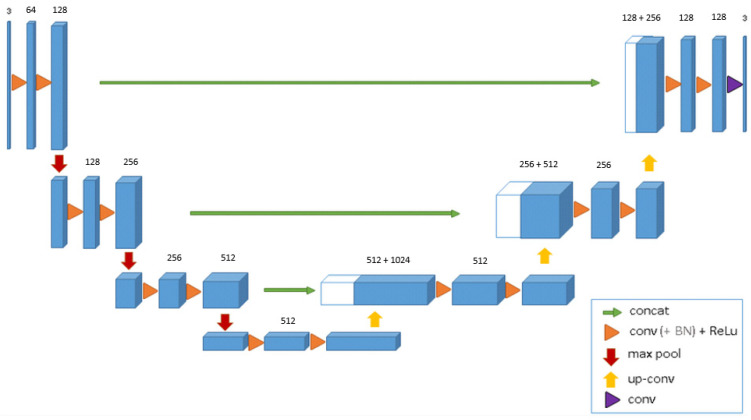
Standard 3D U-Net architecture.

**Figure 4 cancers-16-01130-f004:**

Structure of a Wide Residual Network with width k.

**Figure 5 cancers-16-01130-f005:**
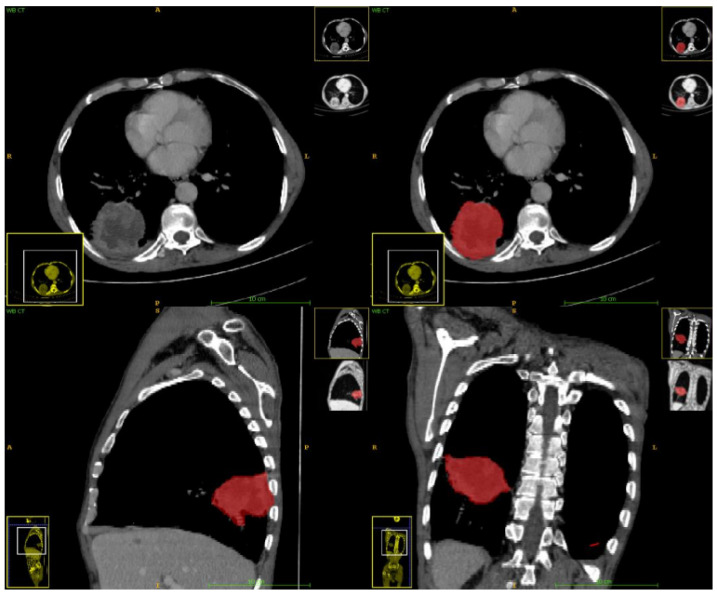
Segmentation of the tumour (in red) done manually in all the three planes using multiplanar reconstruction for the extraction of radiomics features.

**Figure 6 cancers-16-01130-f006:**
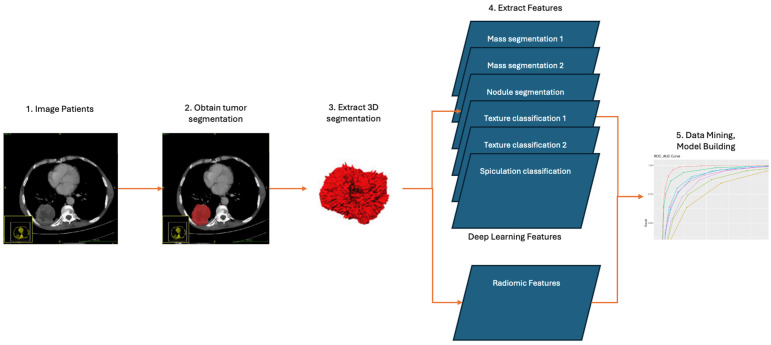
Deep learning features and radiomics features workflow.

**Figure 7 cancers-16-01130-f007:**
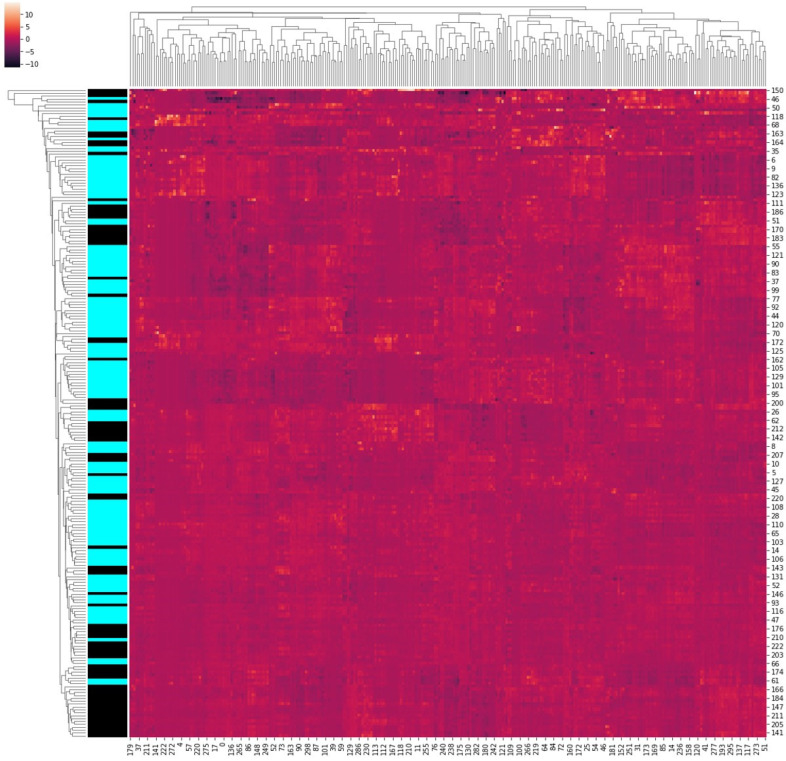
Covariance matrix of radiomic features.

**Figure 8 cancers-16-01130-f008:**
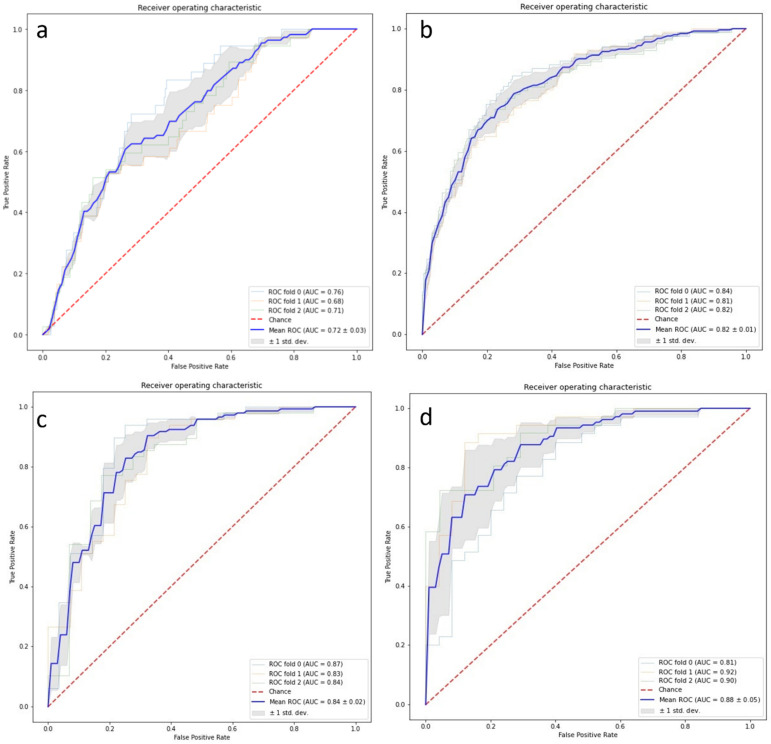
(**a**) ROC curves based on 990 studies with three-fold cross-validation of radiomic features. (**b**) ROC curves based on 990 studies with three-fold cross-validation of deep learning features. (**c**) ROC curves before adding the semantic features to the 223 cases. (**d**) ROC curve after the addition of semantic features to the 223 cases.

**Figure 9 cancers-16-01130-f009:**
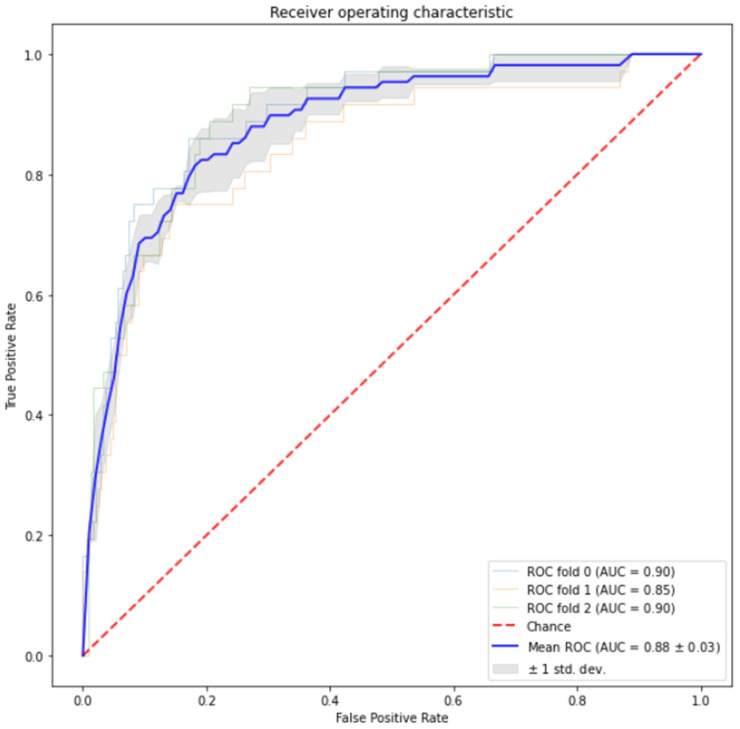
ROC curves from 990 studies with three-fold cross-validation on deep learning features.

**Table 1 cancers-16-01130-t001:** List of 9 feature vectors, the models used and their combinations.

**Feature Vector**	**Model**	**Patch Size**	**Processing**	**Feature Vector Size**
1	Mass segmentation 1	132 × 132 × 132	Original image is at 1× zoom	512
2	Mass segmentation 1	132 × 132 × 132	Original image is at 2× zoom	512
3	Mass segmentation 1	132 × 132 × 132	Original image is at 0.5× zoom	512
4	Mass segmentation 2	132 × 132 × 132	Original image at 0.5× zoom	512
5	Nodule segmentation	132 × 132 × 132	Original image is 0.5× zoom	512
6	Texture classification 1	64 × 64 × 64	Original image is at 0.5× zoom	320
7	Texture classification 1	32 × 32 × 32	Original image is at 0.5× zoom	320
8	Texture classification 2	64 × 64 × 64	Original image is at 0.5× zoom	320
9	Spiculation classification	64 × 64 × 64	Original image is at 0.5× zoom	512

**Table 2 cancers-16-01130-t002:** Top-performing radiomics features in predicting EGFR mutation.

Radiomics Features	Importance	Normalized Importance (%)
Entropy	0.046	71.3
Variance	0.03	56.4
Enhance Count	0.036	57.4
Core Count	0.032	49.70
Cluster Shade	0.033	46.2
Core Count	0.031	44.4
Two-step Cluster Number Based on Age	0.032	42.10
Edema Count	0.030	42.6
Dissimilarity	0.027	46.3
Core Count	0.03	42.8
Difference in Entropy	0.028	41.9
Enhance Count	0.025	42.9
Variance	0.026	38.0
Maximum Probability	0.029	36.9
Sum of Variance	0.027	36.7
Homogeneity	0.026	36.5
Minimum Probability	0.026	35.4
Correlation	0.022	32.6
Inverse Difference	0.024	32.3
Contrast	0.023	29.5
Cluster Shade	0.018	26.6
Correlation	0.017	24.5
Variance	0.017	22.7
Maximum Probability	0.015	19.5
Cluster Prominence	0.016	18.5
Dissimilarity	0.013	18.9
Auto-Correlation	0.015	18.5
Inverse Difference	0.013	18.4
Sum of Squares Variance	0.011	18.2
Difference in Entropy	0.013	17.5
Average	0.015	17.2
Maximum Probability	0.013	16.3
Homogeneity	0.01	15.4
Difference in Entropy	0.008	13.9
Mean	0.008	13.4
Cluster Prominence	0.008	11.2
Sum Average	0.008	11.1
Inverse Difference	0.007	9
Minimum	0.006	7.8
Contrast	0.005	6.4
Sum of Intensities	0.003	6.3
Contrast	0.004	4.7
Homogeneity	0.002	2.5
Contrast	0.002	1.7
Dissimilarity	0.001	1.5

**Table 3 cancers-16-01130-t003:** Association between clinical features and EGFR mutation status.

Sr. No.	Variables		N	EGFR Wild Type	EGFR Mutant Type	Mean Age	*p* Value ^	Univariate OR {CI}
	Total Patients		223	102	121			
1	Median Age (years)			57 (48–62.8)	54 (46–59)		0.095	0.981 {0.95, 1.007}
2	Gender [%]	Male	143	79 [77.55%]	68 [56.25%]	54.7 (28–80)		
		Female	76	23 [22.5%]	53 [43.8%]	52.7 (31–75)	0.001	2.6 {1.48, 4.81}
3	Smoking status [%]	Yes		44 [45.1%]	22 [18.1%]		0.001	3.5 {1.94, 6.50}
		No		57 [55.9%]	99 [81.8%]			
4	Tumour stage [%]	III		12 [11.8%]	1 [0.8%]			
		IV		90 [88.2%]	120 [99.2%]		0.008	16 {2.04, 125.31}

Note—OR = odds ratio; CI—confidence interval. Data in parentheses [] are the percentage and parenthesis () are the range. ^ *p* value was based on a comparison between the EGFR mutation group and the wild-type group. Data in parentheses {} are 95% confidence intervals (CIs).

**Table 4 cancers-16-01130-t004:** CT Features and EGFR mutation status.

	Variables		EGFR Wild Type	EGFR Mutant Type	*p* Value ^	OR (Odds Ratio)
**1**	Tumour size	≤5 CM (Ref.)	60 (58.8)	65 (53.7)	Reference	
		>5 CM	42 (41.2)	56 (46.3)	0.44	1.231 {0.72, 2.09}
**2**	Tumour lobe location	Right upper lobe (Ref.)	25 (24.5)	36 (29.8)	Reference	
		Right middle lobe	7 (6.9)	9 (7.4)	0.84	0.893 {0.29, 2.71}
		Right lower lobe	17 (16.7)	21 (17.4)	0.71	0.858 {0.37, 1.94}
		Left upper lobe	34 (33.3)	30 (24.8)	0.17	0.613 {0.30, 1.24}
		Left lower lobe	19 (18.6)	25 (20.7)	0.82	0.914 {0.41, 2.003}
**3**	Tumour distribution	Central	9 (8.8)	14 (11.6)	0.87	0.929 {0.36, 2.34}
		Peripheral	53 (52.0)	40 (33.1)	**0.01**	0.451 {0.25, 0.79}
		Both (Ref.)	40 (39.2)	67 (55.4)	Reference	
**4**	Contour (%)	Round/oval (Ref.)	0 (0.0)	1 (0.8)	Reference	
		Irregular	99 (97.1)	117 (96.7)	0.87	0.886 {0.19, 4.05}
**5**	Margins	Well defined (Ref.)	25 (24.5)	24 (19.8)	Reference	
		Poorly defined	77 (75.5)	97 (80.2)	0.40	1.312 {0.69, 2.47}
**6**	Spiculations (%)	Absent (Ref.)	28 (27.5)	23 (19.0)	Reference	
		Fine spiculations	38 (37.3)	45 (37.2)	0.30	1.442 {0.71, 2.90}
		Coarse spiculations	36 (35.3)	53 (43.8)	0.10	1.792 {0.89, 3.59}
**7**	Enhancement pattern	Homogeneous (Ref.)	14 (13.7)	13 (10.7)	Reference	
		Mild/moderate heterogeneous	43 (42.2)	41 (33.9)	0.95	1.027 {0.43, 2.44}
		Marked heterogeneous	45 (44.1)	67 (55.4)	0.27	1.603 {0.68, 3.73}
**8**	Enhancement heterogeneity	Maximum Enhancement	60.5 [50.25–75.5]	71 [59–87]	**0.001**	1.024 {1.01, 1.03}
		Minimum Enhancement	35 [28–48]	44 [35–55]	**0.002**	1.026 {1.009, 1.04}
		Average Enhancement A. Average Enhancement < 54 HU B. Average Enhancement > 54 HU	48 [41–60] 64 (62.7) 38(37.3)	57 [48–66] 47 (38.8) 74 (61.2)	**0.004** Reference **<0.001**	2.652 {1.54, 4.55}
		Relative Enhancement to reference artery	0.32 [0.24–0.4]	0.35 [0.28–0.41]	0.116	9.733
**9**	Texture	Predominant solid with associated GGO component	21 (20.6)	11 (9.1)	Reference	
		Pure Solid (no associated ground glass component)	81 (79.4)	109 (90.1)	**0.028**	2.355 {1.09, 5.06}
**10**	Air bronchogram	Absent (Ref.)	65 (63.7)	63 (52.1)		
		Present	37 (36.3)	58 (47.9)	0.08	1.617 {0.94, 2.77}
**11**	Bubble like lucency	Absent (Ref.)	94 (92.2)	110 (90.9)	Reference	
		Present	8 (7.8)	11 (9.1)	0.73	1.175 {0.45, 3.04}
**12**	Cavitation	Absent (Ref.)	99 (97.1)	114 (94.2)	Reference	
		Present	3 (2.9)	7 (5.8)	0.31	2.026 {0.51, 8.04}
**13**	Peripheral emphysema	Absent (Ref.)	85 (83.3)	114 (94.2)	Reference	
		Mild/moderate	14 (13.7)	6 (5.0)	**0.024**	0.320 {0.11, 0.86}
		Marked	3 (2.9)	1 (0.8)	0.23	0.249 {0.025, 2.43}
**14**	Peripheral fibrosis	Absent (Ref.)	67 (65.7)	73 (60.3)	Reference	
		Mild/Moderate	27 (26.5)	38 (31.4)	0.39	1.292 {0.71, 2.34}
		Marked	8 (7.8)	10 (8.3)	0.78	1.147 {0.42, 3.07}
**15**	Fissure attachment	Absent (Ref.)	43 (42.2)	25 (20.7)	Reference	
		Present	59 (57.8)	96 (79.3)	**0.001**	2.799 {1.55, 5.04}
**16**	Pleural attachment	Absent (Ref.)	12 (11.8)	13 (10.7)	Reference	
		Present	90 (88.2)	108 (89.3)	0.80	1.108 {0.48, 2.54}
**17**	Pleural retraction	Absent (Ref.)	38 (37.3)	24 (19.8)	Reference	
		Present	64 (62.7)	97 (80.2)	**0.004**	2.400 {1.31, 4.37}
**18**	Vascular convergence	Absent (Ref.)	101 (99.0)	119 (98.3)	Reference	
		Present	1 (1.0)	2 (1.7)	0.66	1.697 {0.15, 18.99}
**19**	Thickened Broncho vascular bundle	Absent (Ref.)	50 (49.0)	50 (41.3)	Reference	
		Present	52 (51.0)	71 (58.7)	0.25	1.365 {0.80, 2.32}
**20**	Calcifications	Absent (Ref.)	100 (98.0)	115 (95.0)		2.609
		Present	2 (2.0)	6 (5.0)	0.24	{0.51, 13.2}
**21**	Lymphadenopathy	Absent (Ref.)	28 (27.5)	29 (24.0)	Reference	
		Present	74 (72.5)	92 (76.0)	0.55	1.200 {0.65, 2.19}
**22**	Vascular involvement	Absent (Ref.)	56 (54.9)	39 (32.2)	Reference	
		Present	46 (45.1)	82 (67.8)	**0.001**	2.560 {1.48, 4.41}
**23**	Pleural effusion	Absent (Ref.)	71 (69.6)	67 (55.4)	Reference	
		Present ipsilateral	30 (29.4)	51 (42.1)	**0.04**	1.801 {1.02, 3.15}
		Present contralateral	1 (1.0)	3 (2.5)	0.32	3.179 {0.32, 31.32}
**24**	Lymphangitic spread	Absent (Ref.)	76 (74.5)	82 (67.8)	Reference	
		Present	26 (25.5)	39 (32.2)	0.27	1.390 {0.77, 2.49}
**25**	Pleural nodularity	Absent (Ref.)	66 (64.7)	71 (58.7)	Reference	
		Present	34 (33.3)	50 (41.3)	0.35	1.291 {0.74, 2.22}
**26**	Lobulations	Absent (Ref.)	80 (78.4)	98 (81.0)	Reference	
		Present < 3	1 (1.0)	4 (3.3)	0.29	3.265 {0.35, 29.79}
		Present > 3	21 (20.6)	19 (15.7)	0.38	0.739 {0.37, 1.46}
**27**	Tumour lobe metastatic nodule	Absent (Ref.)	39 (38.2)	30 (24.8)	Reference	
		Present	63 (61.8)	91 (75.2)	**0.032**	1.878 {1.05, 3.33}
**28**	Non-tumour lobe metastatic nodule	Absent (Ref.)	42 (41.2)	25 (20.7)	Reference	
		Present	60 (58.8)	96 (79.3)	**0.001**	2.688 {1.48, 4.85}

Note—OR = odds ratio; CI—confidence interval. Data in parentheses [] are the percentage and parenthesis () are the range. ^ *p* value was based on the comparison between the EGFR mutation group and the wild-type group. Data in parentheses {} are 95% confidence intervals (CIs). *p*-values with statistical significance are shown in bold.

## Data Availability

The datasets generated in this study are available on request to the corresponding authors.

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
