# Peer review of "Deep-Learning-Based Predictive Imaging Biomarker Model for EGFR Mutation Status in Non-Small Cell Lung Cancer from CT Imaging"

_cancers, 2024, doi:10.3390/cancers16061130_

Round 1
Reviewer 1 Report
Comments and Suggestions for Authors
I am glad to review the manuscript entitled Deep learning-based predictive imaging biomarker model for EGFR mutation status in non-small cell lung cancer from CT imaging. This manuscript addresses an important and timely topic, leveraging deep learning for predicting EGFR mutation status in non-small cell lung cancer (NSCLC) patients. Although there are some major problems which have to be addressed through a significant revision, the overall quality of the research is satisfactory.
The study's methodology, utilizing a machine learning pipeline and deep convolutional neural networks on CT images, is well-structured and rigorous. The combined radiomics and deep learning model achieving an 88% accuracy is promising for its potential clinical applications. This is a carefully done study and the model provided is of considerable valuable, but there are still some problems in this paper, which could cause difficulties to the readers.
The major problems include:
1. The current reliance solely on area under the curve (AUC) for assessment may not provide a comprehensive view of the model's effectiveness. It is strongly recommended to include a consolidated table summarizing various performance metrics, such as sensitivity, specificity, and AUC. Additionally, the authors are encouraged to consider presenting a confusion matrix, as it can provide further insights into the model's performance in predicting EGFR mutation status.
2. The manuscript indicates that the deep learning model incorporates 1422 features, the radiomic model includes 200 features, and the fusion model comprises 2000 features. The extensive feature set raises concerns about potential overfitting, and it prompts the question of whether the authors have assessed and addressed the risk of overfitting in their models. It is recommended that the authors discuss the possibility of overfitting and consider employing different feature selection methods for further dimensionality reduction.
3. The graphical elements, including figures and tables, lack adherence to standard conventions and aesthetic principles. The tabular representation should adopt a more reader-friendly format, preferably using a standard three-line table for clarity. The image quality throughout the manuscript is a significant issue, with pixelation and blurriness affecting the overall clarity. Specifically, in Figures 6 and 7, the pixelation is prominent, and the axes and labels lack precision. Similar issues are observed in other figures.
The minor problems include:
1. The figures, being crucial components of the article, require refinement to succinctly and clearly convey the central ideas of the study. Specifically, Figures 2 and 3, depicting standard neural network and 3D U-Net architectures, might be too detailed for standalone inclusion. It is recommended to streamline and condense the graphical content, possibly by incorporating a comprehensive research workflow diagram to replace the dispersed process illustrations from Figures 1 to 6.
2. The manuscript mentions the use of 3D U-Net for lesion segmentation. It is crucial to ascertain whether the authors have evaluated the accuracy of the segmentation network. If such an evaluation has been conducted, I suggest including the segmentation network's precision and other relevant metrics in the supplementary materials, along with the learning curves.
3. It is recommended that the authors supplement the presented AUC values with their corresponding 95% confidence intervals.
4.It is suggested to standardize the writing of the manuscript by introducing abbreviations for terms the first time they appear. For instance, on the second page, line 74, the abbreviation "CT" is used, and the full term "Computed Tomography (CT)" is introduced in line 75. Please carefully review the entire manuscript to identify and correct any other similar errors that may have been overlooked.
Thank you!
Comments on the Quality of English Language
The overall quality of the English writing in the manuscript is generally satisfactory, but there are some minor issues that need improvement, such as ensuring consistency in terminology and abbreviations throughout the entire document. I encourage the authors to review the manuscript once again to ensure language clarity and coherence. It is important to confirm that the methods, results, and discussion sections are logically structured and presented in a reader-friendly manner. This revision will contribute to the overall readability and impact of the manuscript.
Author Response
Thank you for your insightful feedback and review of our manuscript.
Major problems:
- The current reliance solely on area under the curve (AUC) for assessment may not provide a comprehensive view of the model's effectiveness. It is strongly recommended to include a consolidated table summarizing various performance metrics, such as sensitivity, specificity, and AUC. Additionally, the authors are encouraged to consider presenting a confusion matrix, as it can provide further insights into the model's performance in predicting EGFR mutation status.
- The AUC plots and score give us an overall idea of the performance of the model at various false positive rates, and the sensitivity corresponding to a specific false positive rate can be determined from the plot. For eg: for a false positive rate of 0.6, we have a sensitivity of about 0.9. Specificity, precision, and confusion matrix can be calculated based on the false positive rate that we choose.
- The manuscript indicates that the deep learning model incorporates 1422 features, the radiomic model includes 200 features, and the fusion model comprises 2000 features. The extensive feature set raises concerns about potential overfitting, and it prompts the question of whether the authors have assessed and addressed the risk of overfitting in their models. It is recommended that the authors discuss the possibility of overfitting and consider employing different feature selection methods for further dimensionality reduction.
- Dimensionality reduction has been elaborated in lines 182-189. The deep learning model actually had 4032 features which came down to 1422 features using correlation coefficients. We identified the correlation between every other feature and removed those features that were redundant.
- The graphical elements, including figures and tables, lack adherence to standard conventions and aesthetic principles. The tabular representation should adopt a more reader-friendly format, preferably using a standard three-line table for clarity. The image quality throughout the manuscript is a significant issue, with pixelation and blurriness affecting the overall clarity. Specifically, in Figures 6 and 7, the pixelation is prominent, and the axes and labels lack precision. Similar issues are observed in other figures.
- Agree, the tables have been modified to a standard three-line table as suggested. The blurred images in question have been corrected.
Minor problems:
- The figures, being crucial components of the article, require refinement to succinctly and clearly convey the central ideas of the study. Specifically, Figures 2 and 3, depicting standard neural network and 3D U-Net architectures, might be too detailed for standalone inclusion. It is recommended to streamline and condense the graphical content, possibly by incorporating a comprehensive research workflow diagram to replace the dispersed process illustrations from Figures 1 to 6.
- After careful consideration, we have reviewed the current figures and their role in presenting the study's findings. While we acknowledge your suggestion to refine Figures 2 and 3, which depict standard neural network and 3D U-Net architectures, we believe that the level of detail provided in these figures is appropriate for the intended audience and contributes significantly to the understanding of our methodologies.
- The manuscript mentions the use of 3D U-Net for lesion segmentation. It is crucial to ascertain whether the authors have evaluated the accuracy of the segmentation network. If such an evaluation has been conducted, I suggest including the segmentation network's precision and other relevant metrics in the supplementary materials, along with the learning curves.
- Metrics of the best model (epoch 22) on the validation dataset -
Sensitivity - 84.12%
Precision - 89.19%
Specificity - 98.88%
Dice score - 0.85
The learning curves have been included as supplementary material.
- It is recommended that the authors supplement the presented AUC values with their corresponding 95% confidence intervals.
- The values are already presented in the graph, e.g. 0.82 +/- 0.01, the 95% confidence interval is [0.81, 0.83]. The same has also been elaborated in the text.
4. It is suggested to standardize the writing of the manuscript by introducing abbreviations for terms the first time they appear. For instance, on the second page, line 74, the abbreviation "CT" is used, and the full term "Computed Tomography (CT)" is introduced in line 75. Please carefully review the entire manuscript to identify and correct any other similar errors that may have been overlooked.
- Appropriate corrections have been made in the manuscript.
The overall quality of the English writing in the manuscript is generally satisfactory, but there are some minor issues that need improvement, such as ensuring consistency in terminology and abbreviations throughout the entire document. I encourage the authors to review the manuscript once again to ensure language clarity and coherence. It is important to confirm that the methods, results, and discussion sections are logically structured and presented in a reader-friendly manner. This revision will contribute to the overall readability and impact of the manuscript.
- We agree with your assessment and have meticulously proofread the manuscript for the above.
Reviewer 2 Report
Comments and Suggestions for Authors
The authors presented an artificial intelligence-based radiomics and DLR model that demonstrated high accuracy in EGFR mutation prediction, serving as a non-invasive and user-friendly imaging biomarker for predicting EGFR mutation status.
1. Initially, the authors write about 990 patients, then they talk about 223 patients with different EGFR mutation status. Why is this so? Section 3.5 mentions 990 studies, and is this about patients?
2. Figures 6-9 are of very poor quality, the inscriptions are small and unreadable. It needs to be redone.
3. Do the authors suggest that determination of EGFR mutant status will be carried out only by CT? or will there be control using traditional histology? If a CT image shows squamous cell carcinoma, what group will it be classified into? Is preliminary confirmation of the histological type of lung adenocarcinoma necessary?
4. What do you think, if we parallel the determination of EGFR mutation status by CT and blood, how much will the accuracy increase and will it be comparable to a biopsy?
Author Response
Thank you for your insightful feedback and review of our manuscript.
- Initially, the authors write about 990 patients, then they talk about 223 patients with different EGFR mutation status. Why is this so? Section 3.5 mentions 990 studies, and is this about patients?
- Yes, it refers to 990 patients in section 3.5. Studies refer to CT studies. The mass segmentation networks were trained on 990 patients. 223 patients were selected for extraction of semantic features.
- Figures 6-9 are of very poor quality, the inscriptions are small and unreadable. It needs to be redone.
- Appropriate changes have been made.
- Do the authors suggest that determination of EGFR mutant status will be carried out only by CT? or will there be control using traditional histology? If a CT image shows squamous cell carcinoma, what group will it be classified into? Is preliminary confirmation of the histological type of lung adenocarcinoma necessary?
- CT determination can be an adjunct but we do not propose replacement of the traditional method of histological determination of EGFR status. CT can be a useful tool to triage patients while the EGFR results are awaited and also in regions where EGFR testing is not available.
- What do you think, if we parallel the determination of EGFR mutation status by CT and blood, how much will the accuracy increase and will it be comparable to a biopsy?
- We believe this would require a separate study to determine the accuracy of both together. However, EGFR testing is a gold standard clinical practice.
Reviewer 3 Report
Comments and Suggestions for Authors
Thank you very much for submitting your important work to the journal. I have several comments.
-First, please add details regarding the two NSCLC trials in which the patient clinical and image data were extracted. If available, please give NCT numbers for the studies and explain these studies shortly.
-The introduction could be shortened. There are several unnecessary comments, like the cellular mechanism of EGFR and the lack of response to EGFR inhibitors in EGFR-mutated lung cancer. Additionally, the most of the references in the introduction was very outdated.
-Please give details regarding the program used for image analysis with the relevant version.
-Please give details regarding the venue for radiomics workflow and DL model construction with relevant versions (phyton, R, etc).
-Most importantly, the discussion is very limited. I recommend the authors give more details regarding the novelty of the findings and the comparison of their findings to the available literature. There are several similar studies in the literature, but none has been cited in the discussion (DOI: 10.3389/fonc.2022.951575, DOI: 10.1038/s41598-024-51630-6, DOI: 10.1183/13993003.00986-2018).
Comments on the Quality of English LanguageThe quality of English is generally acceptable, although there are several typing errors in the paper.
Author Response
Thank you for your insightful feedback and review of our manuscript.
- First, please add details regarding the two NSCLC trials in which the patient clinical and image data were extracted. If available, please give NCT numbers for the studies and explain these studies shortly.
- https://www.ncbi.nlm.nih.gov/pmc/articles/PMC5519810/ (CTRI/2015/08/006113)
- https://pubmed.ncbi.nlm.nih.gov/31411950/ (CTRI/2016/08/007149)
A short explanation of the studies has been included from lines 122-124.
- The introduction could be shortened. There are several unnecessary comments, like the cellular mechanism of EGFR and the lack of response to EGFR inhibitors in EGFR-mutated lung cancer. Additionally, the most of the references in the introduction was very outdated.
- Agree, necessary changes have been incorporated.
- Please give details regarding the program used for image analysis with the relevant version.
- The entire image analysis was performed using Python3.8 framework.
- Please give details regarding the venue for radiomics workflow and DL model construction with relevant versions (phyton, R, etc).
- Radiomics features are calculated using radiomics-v0.1 library from github
The DL model is constructed using the following frameworks
Python 3.8
Pytorch 1.1
- Most importantly, the discussion is very limited. I recommend the authors give more details regarding the novelty of the findings and the comparison of their findings to the available literature. There are several similar studies in the literature, but none has been cited in the discussion (DOI: 10.3389/fonc.2022.951575, DOI: 10.1038/s41598-024-51630-6, DOI: 10.1183/13993003.00986-2018).
- Agree. Appropriate changes have been made.
Reviewer 4 Report
Comments and Suggestions for Authors
This paper developed and validated a deep learning-based radio-genomic model and radiomic signature, to predict the EGFR mutation in patients with NSCLS, and to assess the semantic and clinical features that can contribute in detecting EGFR mutation. I think this paper is acceptable. It’s a practical work, still I have some comments as below:
1. Line 144, convoluted -> convolutional
2. Figure 2, missing word on ‘Fully-connected l’?
3. Line 177, here mentions that “all these models are trained with a learning rate of 1e-4 and a weight decay of 1e-5”, so what optimizer is used here? Is this parameter setting common for medical-related tasks? List some necessary references here please.
4. The words and numbers presented in Figure 6 and Figure 7 are a bit blurred.
5. More advanced techniques should be reviewed or compared such as: Improved differentiable architecture search with progressive partial channel connections based on channel attention, Evolving deep convolutional neural networks for image classification, Neural architecture transfer.
6. There are many articles related to deep learning medical tasks in the recent two years, I suggest bringing relevant references in this paper.
Comments on the Quality of English LanguageThis paper developed and validated a deep learning-based radio-genomic model and radiomic signature, to predict the EGFR mutation in patients with NSCLS, and to assess the semantic and clinical features that can contribute in detecting EGFR mutation. I think this paper is acceptable. It’s a practical work, still I have some comments as below:
1. Line 144, convoluted -> convolutional
2. Figure 2, missing word on ‘Fully-connected l’?
3. Line 177, here mentions that “all these models are trained with a learning rate of 1e-4 and a weight decay of 1e-5”, so what optimizer is used here? Is this parameter setting common for medical-related tasks? List some necessary references here please.
4. The words and numbers presented in Figure 6 and Figure 7 are a bit blurred.
5. More advanced techniques should be reviewed or compared such as: Improved differentiable architecture search with progressive partial channel connections based on channel attention, Evolving deep convolutional neural networks for image classification, Neural architecture transfer.
6. There are many articles related to deep learning medical tasks in the recent two years, I suggest bringing relevant references in this paper.
Author Response
Thank you for your insightful feedback and review of our manuscript.
- Line 144, convoluted -> convolutional
- Appropriate changes have been made.
- Figure 2, missing word on ‘Fully-connected l’?
- Thank you for the feedback. It should read as fully connected “layer”. The revised image has been added.
- Line 177, here mentions that “all these models are trained with a learning rate of 1e-4 and a weight decay of 1e-5”, so what optimizer is used here? Is this parameter setting common for medical-related tasks? List some necessary references here please.
- Adam optimizer is used. This parameter setting is based on various experiments on parameters and this setting resulted in a better training of the model.
- The words and numbers presented in Figure 6 and Figure 7 are a bit blurred.
- Agree and revised images have been added.
- More advanced techniques should be reviewed or compared such as: Improved differentiable architecture search with progressive partial channel connections based on channel attention, Evolving deep convolutional neural networks for image classification, Neural architecture transfer.
- While we acknowledge your suggestions, these would warrant a separate study, and as such it is not feasible for us to incorporate it into the present manuscript given the timeframe allocated for revisions.
- There are many articles related to deep learning medical tasks in the recent two years, I suggest bringing relevant references in this paper.
- Agree. This has been addressed in the revised discussion.
Reviewer 5 Report
Comments and Suggestions for Authors
Mahajan et al. assessed the capability of radiomics and deep learning approaches to predict EGFR mutation from CT images in patients with adenocarcinoma of the lung. Also the contribute of adding semantic radiologic signatures extracted from the CT in a subset of the patients sample is shown.
The paper is intersting and contributing to the field.
Minor flaws
The architecture of the paper could be improved by better outlining the three information that are being evaluated: namely 1) the radiomics features that can also be extracted with simple manual ROI; 2) the deep learning method that can be used to segmentate and characterize the LC with/out inclusion of radiomics analysis and 3) the radiologic signatures visually evaluated by the radiologist.
Fig.1 A legend is incorrect. From the text it appears to show the AUROC of the radiomic features and not of the deep learning features.
Comments on the Quality of English LanguageEnglish requires some revision
Author Response
Thank you for your insightful feedback and review of our manuscript.
- The architecture of the paper could be improved by better outlining the three information that are being evaluated: namely 1) the radiomics features that can also be extracted with simple manual ROI; 2) the deep learning method that can be used to segmentate and characterize the LC with/out inclusion of radiomics analysis and 3) the radiologic signatures visually evaluated by the radiologist.
This has been discussed in the discussion. However, splitting the discussion into categories wasn’t done.
- Fig.1 A legend is incorrect. From the text it appears to show the AUROC of the radiomic features and not of the deep learning features.
- Agree, changes are made.
English requires some revision
- The manuscript has been proofread for grammatical errors.
Round 2
Reviewer 2 Report
Comments and Suggestions for Authors
I have no further comments on the manuscript.
Reviewer 3 Report
Comments and Suggestions for Authors
Thank you very much conducting the revisions.
Comments on the Quality of English LanguageThe quality of english is acceptable.